

# Effect of *AOX1* and *GAP* transcriptional terminators on transcript levels of both the heterologous and the *GAPDH* genes and the extracellular $Y_{p/x}$ in *GAP* promoter-based *Komagataella phaffii* strains

José M. Viader-Salvadó\*, Nancy Pentón-Piña\*, Yanelis Robainas-del-Pino, José A. Fuentes-Garibay and Martha Guerrero-Olazarán

Instituto de Biotecnología, Facultad de Ciencias Biológicas, Universidad Autónoma de Nuevo León, San Nicolás de los Garza, Nuevo León, Mexico
\* These authors contributed equally to this work.

Corresponding authors
José M. Viader-Salvadó,
jose.viadersl@uanl.edu.mx
Martha Guerrero-Olazarán,
martha.guerrerool@uanl.edu.mx

## ABSTRACT

The constitutive and strong *GAP* promoter ($P_{GAP}$) from the glyceraldehyde-3-phosphate dehydrogenase (*GAPDH*) gene has emerged as a suitable option for protein production in methanol-free *Komagataella phaffii* (syn. *Pichia pastoris*) expression systems. Nevertheless, the effect of the transcriptional terminator from the alcohol oxidase 1 gene ($T_{AOX1}$) or *GAPDH* gene ($T_{GAP}$) within the heterologous gene structure on the transcriptional activity in a $P_{GAP}$-based strain and the impact on the extracellular product/biomass yield ($Y_{p/x}$) has not yet been fully characterized. In this study, we engineered two *K. phaffii* strains, each harboring a single copy of a different combination of regulatory DNA elements (*i.e.*, $P_{GAP}$-$T_{AOX1}$ or $P_{GAP}$-$T_{GAP}$ pairs) within the heterologous gene structure. Moreover, we assessed the impact of the regulatory element combinations, along with the carbon source (glucose or glycerol) and the stage of cell growth, on the transcript levels of the reporter gene and the endogenous *GAPDH* gene in the yeast cells, as well as the extracellular $Y_{p/x}$ values. The results indicate that the regulation of transcription for both heterologous and endogenous *GAPDH* genes, the extracellular $Y_{p/x}$ values, and translation and/or heterologous protein secretion were influenced by the $P_{GAP}$-transcriptional terminator combination, with the carbon source and the stage of cell growth acting as modulatory factors. The highest transcript levels for the heterologous and endogenous *GAPDH* genes were observed in glucose cultures at a high specific growth rate ($0.253\ \text{h}^{-1}$). Extracellular $Y_{p/x}$ values showed an increasing trend as the culture progressed, with the highest values observed in glucose cultures, and in the $P_{GAP}$-$T_{AOX1}$-based strain. The presence of $T_{AOX1}$ or $T_{GAP}$ within the heterologous gene structure activated distinct gene regulatory elements in each strain, leading to differential modulation of gene regulation for the heterologous and the *GAPDH* genes, even though both genes were under the control of the same promoter ($P_{GAP}$). $T_{AOX1}$ induced competitive regulation of transcriptional activity between the two genes, resulting in enhanced transcriptional activity of the *GAPDH* gene. Moreover, $T_{AOX1}$ led to increased mRNA stability and triggered distinct metabolic downregulation mechanisms due to carbon source depletion compared to $T_{GAP}$. $T_{AOX1}$ enhanced translation and/or heterologous protein secretion activity at a high

specific growth rate (0.253 h$^{-1}$), while T$_{GAP}$ was more effective in enhancing post-transcriptional activity at a low specific growth rate (0.030 h$^{-1}$), regardless of the carbon source. The highest extracellular Y$_{p/x}$ was obtained with the P$_{GAP}$-T$_{AOX1}$-based strain when the culture was carried out at a low specific growth rate (0.030 h$^{-1}$) using glucose as the carbon source. The optimization of regulatory elements and growth conditions presents opportunities for enhancing the production of biomolecules of interest.

## INTRODUCTION

The methylotrophic yeast *Komagataella phaffii* (formerly known as *Pichia pastoris*) is considered one of the most useful hosts for producing recombinant proteins, because it can produce significant amounts of extracellular heterologous proteins in a controlled bioprocess using simple defined media and posttranslational modifications are possible (*Cereghino & Cregg, 2000*). Given the typical preference for strong and controllable promoters in heterologous protein production (*Vogl & Glieder, 2013*), the alcohol oxidase 1 gene's promoter (P$_{AOX1}$) is the most used promoter in the *K. phaffii* expression system, because it is a strong promoter tightly regulated using methanol as the inductor. Therefore, we demonstrated the ability of *K. phaffii* to produce and secrete the beta-propeller phytase called FTEII in an active form using a P$_{AOX1}$-based expression system in a benchtop bioreactor (*Viader-Salvadó et al., 2013*). Nevertheless, the use of methanol in large-scale bioreactors has severe problems related to its safe handling (*Mattanovich et al., 2014*). The constitutive and strong *GAP* promoter (P$_{GAP}$) from the glyceraldehyde-3-phosphate dehydrogenase (*GAPDH*) gene has emerged as a suitable option for protein production in methanol-free *K. phaffii* expression systems (*Looser et al., 2015*; *García-Ortega et al., 2019*). GAPDH is a key enzyme in both the glycolytic and gluconeogenesis pathways. Specifically, it catalyzes the conversion of glyceraldehyde-3-phosphate to 1,3-bisphosphoglycerate in a reversible reaction. Glucose is metabolized in the glycolytic pathway to produce pyruvate, along with the generation of ATP and NADH. GAPDH is also involved in glycerol metabolism, which enters the glycolytic pathway after its conversion into dihydroxyacetone phosphate and isomerization to glyceraldehyde-3-phosphate, which is then processed by GAPDH, similar to when glucose is the carbon source (*Çalık et al., 2015*, *Tomàs-Gamisans et al., 2019*). Thus, both carbon sources activate the *GAPDH* gene.

We recently tested a P$_{GAP}$-driven system to extracellularly produce the phytase FTEII (*Herrera-Estala et al., 2022*). As other authors (*García-Ortega et al., 2019*) have mentioned, we realized that the process strategies for protein production with a P$_{GAP}$-driven system are easier than strategies with a P$_{AOX1}$-driven system, because they only involve a batch phase followed by a fed-batch phase using glycerol or glucose.

Despite the currently frequent use of P$_{GAP}$-based systems for protein production, the effect of environmental and molecular factors on protein production outcomes has yet to

be comprehensively evaluated. Although glucose is recommended as a carbon source, since $P_{GAP}$ transcriptional activity is higher in cells grown in glucose than in glycerol (*Waterham et al., 1997*; *Cereghino & Cregg, 2000*), glycerol is suggested to be the best substrate for a batch phase and glucose for the fed-batch phase (*García-Ortega et al., 2013*). Nevertheless, the use of glycerol and glucose for the batch and fed-batch phases, respectively, only increased the volumetric productivity ($Q_p$) by 3% and decreased the production yield derived from the substrate ($Y_{p/s}$) by 1%, compared to the use of glycerol-glycerol (*García-Ortega et al., 2013*). Thus, similar production performance has been achieved using either of these substrates. While some authors recommend using a high specific growth rate ($\mu$) to obtain high protein production levels (*García-Ortega et al., 2013*; *Looser et al., 2015*), other authors recommended a low $\mu$ to increase protein secretion, which in turn, increases extracellular $Q_p$ (*Herrera-Estala et al., 2022*).

Transcriptional terminators are another important factor that impacts heterologous gene expression since besides their main function in transcriptional termination, the 3'-untranslated region (3'UTR) harbored within the terminator sequence also influences mRNA stability as well as transcriptional and translational efficiencies (*Mayr, 2019*; *Kuersten & Goodwin, 2003*). Therefore, the combination of terminators with an appropriate promoter has proved to be an effective strategy for tuning gene expression (*Curran et al., 2013*; *Vogl et al., 2016*; *Ramakrishnan et al., 2020*; *Ito et al., 2020*; *Robainas-del-Pino et al., 2023*). In any case, knowledge regarding the functionality of *K. phaffii* terminators is still limited (*Ito et al., 2020*). Specifically, the effect of the transcriptional terminator from the alcohol oxidase 1 gene ($T_{AOX1}$) or the *GAP* transcriptional terminator ($T_{GAP}$) from the *GAPDH* gene in the heterologous gene structure on the transcriptional activity in a $P_{GAP}$-based strain is unknown, and the impact on the extracellular product/biomass yield ($Y_{p/x}$) has not been fully characterized. This is mainly due to the commercially available $P_{GAP}$-based vectors only harbor the $T_{AOX1}$.

In this work, we identified a putative $T_{GAP}$ sequence and constructed two *K. phaffii* strains, each harboring a single copy of a different combination of the regulatory DNA elements ($P_{GAP}$-$T_{AOX1}$ or $P_{GAP}$-$T_{GAP}$) within the heterologous gene structure to extracellularly produce the phytase FTEII. Moreover, we assessed the impact of the regulatory element combinations, along with the carbon source (glucose or glycerol) and the stage of cell growth, on the transcript levels of the heterologous *FTEII* and the endogenous *GAPDH* genes in the yeast cells. We also assessed the extracellular $Y_{p/x}$ values. The results indicate that the regulation of transcription for both heterologous and endogenous *GAPDH* genes, extracellular $Y_{p/x}$ values, and the translation and/or heterologous protein secretion were influenced by the $P_{GAP}$-transcriptional terminator combination, with the carbon source and the stage of cell growth acting as modulators.

## MATERIALS AND METHODS

### Strains, plasmids, media, enzymes, chemicals

The strain *Komagataella phaffii* KM71 (*his4*) was obtained from Thermo Fisher Scientific (Waltham, MA, USA). The cloning vector, pUCIDT-AMP, was sourced from Integrated DNA Technologies, Inc. (Coralville, IA, USA). The *K. phaffii* KM71GAHFTEII strain and

the plasmid pP$_{GAP}$-FTEII-T$_{AOX1}$ (formerly called pGAHFTEII) were constructed earlier in our laboratory (*Herrera-Estala et al., 2022*). This plasmid, derived from the vector pPIC9 (Thermo Fisher Scientific, Waltham, MA, USA), contains an expression cassette that includes the P$_{GAP}$ sequence, followed by the *Saccharomyces cerevisiae* alpha-factor prepro-secretion signal coding sequence, and a nucleotide sequence encoding the mature beta-propeller phytase FTEII, optimized with preferred codons (*Viader-Salvadó et al., 2010*). It also includes the T$_{AOX1}$ sequence (nucleotides 240823 to 241156 from the *K. phaffii* CBS 7435 chromosome 4, GenBank accession number FR839631.1), along with a functional copy of the histidinol dehydrogenase (*HIS4*) gene to restore the auxotrophy of the host cells. Enzymes such as Q5 Hot Start High-Fidelity DNA polymerase, Endo Hf glycosidase, and the restriction endonucleases *Bsu*36I and *Not*I were obtained from New England Biolabs (Beverly, MA, USA). *Sal*I restriction enzyme was acquired from Clontech (Palo Alto, CA, USA). Additional enzymes, including M-MLV reverse transcriptase, RQ1 RNase-free DNase, and GoTaq DNA polymerase, and the oligo(dT)$_{15}$ primer were purchased from Promega (Madison, WI, USA). SCRIPT reverse transcriptase, the oligo(dT)$_{20}$ primer, and the qPCR SybrMaster mix were sourced from Jena Bioscience GmbH (Jena, Germany). Other oligonucleotides and PrimeTime qPCR Probes were from Integrated DNA Technologies, Inc. (Coralville, IA, USA); the sequences are shown in Table S1. RNAlater solution was from Ambion (Grand Island, NY, USA). Yeast extract-peptone-dextrose (YPD), regeneration dextrose base (RDB), and buffered minimal glycerol (BMG) media were prepared following the instructions provided in the *Pichia* expression kit (Thermo Fisher Scientific, Waltham, MA, USA). Modifications to the standard BMG medium, labeled as BMGlc and BMGly media, contained 30 mM glucose (0.54 % [w/v]) or 30 mM glycerol (0.28 % [w/v]), instead of the usual 1 % (w/v) glycerol and also were supplemented with 0.1 % (w/v) CaCl$_2$. All other chemicals were acquired from Sigma-Aldrich Co. (St. Louis, MO, USA) or Productos Químicos Monterrey (Monterrey, Nuevo León, Mexico).

### *GAPDH* transcriptional terminator (T$_{GAP}$) sequence

To determine the T$_{GAP}$ sequence, data from five in-house RNA-seq analyses of a *K. phaffii* KM71 strain grown in glycerol or methanol, available in the NCBI Sequence Read Archive (SRA) under the BioProject accession number PRJNA930494, were mapped to the inter-gene coding sequence (CDS) region of the *GAPDH* gene and the downstream gene (*i.e.*, *NAB6*), using the HISAT2 v2.1.0 program (*Kim, Langmead & Salzberg, 2015*) (Galaxy version 2.1.0) to detect a subregion within the inter-CDS region that lacked aligned RNA reads. Moreover, the 3′UTR sequence of the *GAPDH* gene was determined by the 3′ rapid amplification of cDNA ends (3′RACE) technique (*Frohman, Dush & Martin, 1988*). Total RNA was obtained from 12.5 mg glass beds KM71GAHFTEII-lysed dry cells using the SV Total RNA isolation system (Promega, Madison, WI, USA) according to the manufacturer's instructions. DNA impurities were removed by treatment with RQ1 RNase-free DNase. Cells were previously grown for 3 h in a glycerol-fed batch culture at a μ of 0.054 h$^{-1}$, 24 °C, and pH 6.0, as described elsewhere (*Herrera-Estala et al., 2022*). cDNA synthesis was performed using the M-MLV reverse transcriptase and the T17AP

primer, following the protocols provided by the supplier. The cDNA was amplified by PCR using the RACEAP and the *GAPDH* gene-specific 5qGAP primers and the Q5 Hot Start High-fidelity DNA polymerase. The amplified cDNA product was sequenced at the Instituto de Fisiología Celular of the Universidad Nacional Autónoma de México (UNAM) using the 5qGAP primer. The sequence from the 3′ end of the *GAPDH* coding region to 50 nucleotides downstream of the 3′UTR was considered the *GAPDH* transcriptional terminator ($T_{GAP}$) sequence. The $T_{GAP}$ sequence underwent *in silico* analysis to identify A- and T-rich regions known for their frequent presence downstream of the stop codon in yeast genes (*van Helden, del Olmo & Pérez-Ortín, 2000*).

## Construction of *K. phaffii* $P_{GAP}$-$T_{AOX1}$-based and $P_{GAP}$-$T_{GAP}$-based strains

A DNA fragment containing a *Not*I restriction site, 11 spacer nucleotides, the $T_{GAP}$ sequence, and 507 nucleotides from the 3′ end of the $T_{AOX1}$ to the *Bsu*36I site of the pPIC9 vector, was synthesized, inserted into the pUCIDT-AMP plasmid, and sequenced by Integrated DNA Technologies, Inc. This process generated the $pUCIDT_{GAP}$ plasmid. The fragment released from *Not*I and *Bsu*36I digestion of $pUCIDT_{GAP}$ was ligated into $pP_{GAP}$-FTEII-$T_{AOX1}$ plasmid, which had been previously digested with the same endonucleases, to form the $pP_{GAP}$-FTEII-$T_{GAP}$ expression vector. The accuracy of this construction was confirmed by PCR analysis using FTE1 and 3TH primers directed to the *FTEII* gene and a specific region downstream of the $T_{GAP}$ sequence within the linearized vector, and by DNA sequencing. All DNA manipulations followed standardized protocols (*Green & Sambrook, 2012*). *Komagataella phaffii* cells were transformed with *Sal*I-linearized $pP_{GAP}$-FTEII-$T_{AOX1}$ or $pP_{GAP}$-FTEII-$T_{GAP}$ DNA *via* electroporation as described previously (*Lin-Cereghino et al., 2005*). Transformants were identified based on their ability to grow without histidine on RDB agar plates at 30 °C. To verify the integration of the expression cassette into the *K. phaffii*'s *HIS4* locus, randomly selected colonies from both transformations were analyzed by PCR using two primer pairs (GAPF/FTE2 and FTE1/3TH). The former primer pair targeted the $P_{GAP}$ and the *FTEII* gene, and the second primer pair was directed to the *FTEII* gene and a region downstream of the $T_{AOX1}$ or $T_{GAP}$ terminator sequence. The constructed strains (KM71/$P_{GAP}$-FTEII-$T_{AOX1}$ and KM71/$P_{GAP}$-FTEII-$T_{GAP}$) were hereinafter referred to as $P_{GAP}$-$T_{AOX1}$-based and $P_{GAP}$-$T_{GAP}$-based strains.

## Selection of single-copy transformants from the two constructed strains

Eighteen and twenty-five transformants of each constructed strain were first grown in YPD medium, followed by culturing in BMG medium with 0.1 % (w/v) $CaCl_2$ for 24 h at 30 °C and 250 rpm, with glycerol added after 14 h of incubation to achieve a final concentration of 1 % (w/v), as described previously (*Robainas-del-Pino et al., 2023*). The biomass concentrations in grams of dry cell weight (DCW) per liter were estimated from the optical density at 600 nm ($OD_{600}$), as described by *Robainas-del-Pino et al. (2023)*. The supernatants were separated by centrifugation (16,000 × g, 10 min, 4 °C), and the

extracellular protein/biomass yield was calculated as the ratio of extracellular protein concentration to biomass concentration. All protein concentrations were measured using the Bradford protein assay (*Bradford, 1976*), with bovine serum albumin as the standard.

The three His$^+$ clones from each strain that demonstrated the lowest extracellular protein/biomass yield were selected and analyzed by quantitative PCR (qPCR) to assess the copy number of the expression cassette integrated into the yeast genome. This analysis was conducted using the Mx3005P system (Agilent Technologies, Santa Clara, CA, USA) and PrimeTime qPCR probe assays containing a specific primer pair and a specific hydrolysis probe directed to the heterologous *FTEII* or endogenous *GAPDH* coding sequences, as described earlier (*Herrera-Estala et al., 2022*). One transformant of each strain, identified to possess a single copy of the expression cassette, was selected for further experiments.

### Functionality analysis of the $P_{GAP}$-$T_{AOX1}$ and $P_{GAP}$-$T_{GAP}$ combinations

The functionality of the two combinations of cis-regulatory elements (*i.e.*, $P_{GAP}$-$T_{AOX1}$ and $P_{GAP}$-$T_{GAP}$) on the heterologous gene structure was confirmed through the detection of *FTEII* transcripts *via* reverse transcription-PCR (RT-PCR), along with the analysis of FTEII protein *via* SDS-polyacrylamide gel electrophoresis (SDS-PAGE) and measurement of phytase activity. $P_{GAP}$-$T_{AOX1}$-based and $P_{GAP}$-$T_{GAP}$-based cells were grown in BMG medium with 0.1 % (w/v) $CaCl_2$, for 10 h at 30 °C and 250 rpm. Total RNA was obtained as above. Two-step RT-PCR assays were performed using M-MLV reverse transcriptase, GoTaq DNA polymerase, oligo(dT)$_{15}$, and the primer pair FTE1/FTE2 as described previously (*Robainas-del-Pino et al., 2023*). Negative RT-PCR and PCR controls, along with a positive RT-PCR control for detecting the β-actin transcripts using a specific primer pair (5ACT/3ACT) were also included in the RT-PCR assays.

The supernatant from the 24 h BMG culture of each strain was concentrated 100-fold to a protein level of 2 mg/mL and desalted using 10-kDa Amicon Ultra-4 filters (Millipore, Burlington, MA, USA) at 4 °C. The concentrated and desalted samples were treated with Endo Hf for 1 h at 37 °C, and subjected to SDS-PAGE on a 12% Coomassie blue-stained gel for protein analysis.

The two supernatants were also buffer-exchanged using a PD-10 column (GE Healthcare Bio-Sciences Corp, Piscataway, NJ, USA), and the volumetric extracellular phytase activity was quantified by measuring the phosphate liberation from 5 mM sodium phytate (*Viader-Salvadó et al., 2010*). One unit of phytase activity was considered as the quantity of enzyme needed to release 1 μmol of phosphate in 1 min from sodium phytate at pH 7.5 and 37 °C.

### Growth kinetics, transcript levels, and extracellular $Y_{p/x}$

The two single-copy strains were grown in BMGlc and BMGly media (initial $OD_{600}$ of 0.2) for 24 h at 30 °C and 250 rpm. Samples were collected at five culture times (3, 6, 12, 18, and 24 h) to assess the growth by $OD_{600}$; the biomass concentrations were estimated as above. Specific growth rates (μ) at the exponential growth phase (3 to 12 h) were estimated from the slope of the natural log-linear plot of biomass concentration *vs.* culture time. The biomass concentration *vs.* culture time data was also fitted to the integration solution of the

Verhulst-Pearl logistic equation (*López et al., 2010*) by minimizing the residual sum of squares between the integrated equation and the biomass concentrations, using the Solver tool of Microsoft Excel (Microsoft Co., Redmond, WA, USA). The maximum biomass concentrations ($X_{max}$) were obtained directly from the fitted equations. Discrete values of μ at 6, 12, and 18 h of culture were estimated from the ratio of the first derivative of the fitted growth cell curves and the biomass concentration at each time as described previously (*Cos et al., 2005*) and considering the sampling volumes to be negligible.

The activities of the regulatory element combinations ($P_{GAP}$-$T_{AOX1}$ and $P_{GAP}$-$T_{GAP}$) on the expression of heterologous *FTEII* and endogenous *GAPDH* genes in each single-copy strain were determined by the reverse transcription-quantitative PCR (RT-qPCR) using the same thermocycler that was employed for gene copy number determination. *YPT1* gene was used as the normalizer. Cells from 6, 12, and 18 h of culture were collected by centrifugation as above, and preserved in RNAlater solution until further RNA isolation as described above. The cDNA synthesis was performed from 1 μg RNA using the SCRIPT cDNA synthesis kit and oligo $(dT)_{20}$, following the manufacturer's instructions. *FTEII* and *GAPDH* cDNAs from each sample were amplified in duplicate using the PrimeTime qPCR probe assays as described above for gene copy number determination. *YPT1* cDNA was amplified using the qPCR SybrMaster mix and the 5qYPT1 and 3qYPT1 primers as described earlier (*Caballero-Pérez et al., 2021*). The *FTEII* and *GAPDH* transcript levels were calculated as fold changes to the *YPT1* transcript level of the same sample using a previously determined calibration curve with genomic DNA for each gene-specific cDNA amplification.

Moreover, the supernatants at each sampling time were concentrated, and diafiltrated by ultrafiltration as described previously (*Robainas-del-Pino et al., 2023*). The extracellular $Y_{p/x}$ was estimated as the ratio of volumetric extracellular phytase activity to the increase in biomass concentration from the beginning of the culture.

Ratios of *FTEII*-transcript levels and extracellular $Y_{p/x}$ ratios were calculated by comparing values obtained from cultures of the $P_{GAP}$-$T_{AOX1}$-based strain to those from the $P_{GAP}$-$T_{GAP}$-based strain using the same carbon source (inter-strain ratios). These two ratios were also calculated by comparing values from cultures of the same strain using glucose to those using glycerol (carbon-source ratios). The ratios of the *FTEII*-transcript levels and the extracellular $Y_{p/x}$ ratios were then compared to elucidate the impact of the transcriptional terminator within the heterologous gene structure and the carbon source on post-transcriptional events, such as translation and/or heterologous protein secretion.

The μ values, transcript levels, extracellular $Y_{p/x}$ values, *FTEII*-transcript levels ratios, and extracellular $Y_{p/x}$ ratios were statistically compared using a Student's t-test with a significance level of 0.05.

# RESULTS

## $T_{GAP}$ sequence

The RNA mapping analysis of the inter-CDS region of the *GAPDH* gene and the downstream gene showed the presence of RNA reads throughout the full length of the

inter-CDS region. Therefore, to determine the TGAP sequence, the 3′RACE analysis was more conclusive. After DNA sequencing of the amplified product from the 3′RACE assay, 51 nucleotides downstream of the *GAPDH* coding region were identified as the *GAPDH* 3′UTR sequence, ending with a TA, which is characteristic of a yeast RNA cleavage and polyadenylation site (*Guo & Sherman, 1996*; *Graber, McAllister & Smith, 2002*). This sequence had 100 % identity with the 51 nucleotides downstream of the *GAPDH/TDH3* CDS described for the *K. phaffii* CBS 7435 chromosome 2 (GenBank accession number FR839629.1). Therefore, the $T_{GAP}$ sequence used in this work had a full length of 101 nucleotides (*i.e.*, nucleotides 1587005 to 1587105 from *K. phaffii* CBS 7435 chromosome 2) (Fig. S1). Moreover, four sequences rich in adenine and thymine (GTATGT, AAATAG, TTCATT, and TATCTA) commonly found downstream of CDSs in yeasts (*van Helden, del Olmo & Pérez-Ortín, 2000*) were identified in the $T_{GAP}$ sequence (Fig. S1). These A+T-rich sequences could be the 3′-processing elements e1, e2, e3, and e4 for the $T_{GAP}$ (*Graber, 2003*), also known as efficiency, positioning, near-upstream, and near-downstream elements, respectively (*Graber, McAllister & Smith, 2002*).

## Construction and selection of strains

The construction of the vector $pP_{GAP}$-FTEII-$T_{GAP}$ included a DNA sequence encoding the mature phytase FTEII as the reporter gene. This sequence was in-frame with the alpha-factor prepro-secretion signal coding sequence and located between $P_{GAP}$ and $T_{GAP}$, similar to the previously constructed vector $pP_{GAP}$-FTEII-$T_{AOX1}$ (*Herrera-Estala et al., 2022*), but harboring the $T_{GAP}$ sequence instead of the $T_{AOX1}$ sequence (Fig. S2). PCR analysis of the genomic DNA isolated from $P_{GAP}$-$T_{AOX1}$-based and $P_{GAP}$-$T_{GAP}$-based strains showed the expected 1,311 and 1,047 bp or 1,311 and 814 bp band patterns for the construction harboring the $P_{GAP}$-$T_{AOX1}$ or the $P_{GAP}$-$T_{GAP}$, respectively, using the two primer pairs targeting the $P_{GAP}$ and the *FTEII* gene or the *FTEII* gene and a region downstream of the transcriptional terminator sequence (Fig. S3A). These results confirmed the integration of every expression cassette into the *K. phaffii* genome.

The biomass concentrations at 24-h cultures from 18 and 25 recombinant clones of each strain ($P_{GAP}$-$T_{AOX1}$-based and $P_{GAP}$-$T_{GAP}$-based, respectively) ranged from 7.22 to 9.30 and 7.39 to 9.23 g DCW/L. Moreover, the extracellular protein concentrations ranged between 7.46 to 57.90 and 13.56 to 50.8 mg/L, and the protein/biomass yield ranged from 0.9 to 6.8 and 1.8 to 6.5 mg/g, respectively. The strain selected from each construction showcased the lowest extracellular protein/biomass yield. Subsequent qPCR analysis from the selected strains confirmed a single copy of the heterologous gene within the yeast genome.

## Functionality of the $P_{GAP}$-$T_{AOX1}$ and $P_{GAP}$-$T_{GAP}$ combinations

The RT-PCR analysis for $P_{GAP}$-$T_{AOX1}$-based and $P_{GAP}$-$T_{GAP}$-based strains exhibited the expected 536-bp band corresponding to an *FTEII*-transcript fragment (Fig. S3B).

The SDS-PAGE of the cell-free supernatant from these cultures revealed the typical smear above 39 kDa for phytase FTEII (Fig. S3C, lanes 1 and 3). Upon N-deglycosylation

by Endo Hf, this smear shifted to a 39-kDa band (Fig. S3C, lanes 2 and 4) that corresponds with the predicted molecular mass of phytase FTEII from its amino acid sequence. This outcome substantiates the secretion of phytase FTEII as a highly N-glycosylated protein, as previously observed (*Viader-Salvadó et al., 2010*). Moreover, the supernatant from the cultures of the $P_{GAP}$-$T_{AOX1}$-based and $P_{GAP}$-$T_{GAP}$-based strains exhibited phytase activities of 0.34 and 0.25 U/mL, respectively.

The combined results from the detection of *FTEII* transcripts and FTEII protein *via* RT-PCR, SDS-PAGE, and phytase enzyme activity corroborated the proper functionality of the regulatory sequence combinations (*i.e.*, $P_{GAP}$-$T_{AOX1}$ and $P_{GAP}$-$T_{GAP}$).

## Growth kinetics

The growth kinetics for the two strains ($P_{GAP}$-$T_{AOX1}$-based and $P_{GAP}$-$T_{GAP}$-based), using glucose or glycerol as carbon sources (Fig. 1A), showed the characteristic sigmoid (logistic) population growth. Biomass concentrations exhibited exponential growth from 3 to 12 h of culture with $\mu$ values ranging from $0.213 \pm 0.023$ to $0.249 \pm 0.013$ h$^{-1}$. A higher $X_{max}$ was achieved in cultures grown with glucose compared to those using glycerol (1.6 and 1.8 times higher for the $P_{GAP}$-$T_{AOX1}$-based and $P_{GAP}$-$T_{GAP}$-based strains, respectively), as expected since all the cultures were equimolar in the carbon source, and a molecule of glucose has six carbons whereas a molecule of glycerol has three. Moreover, glucose is a glycolytic carbon source, while glycerol is a gluconeogenic carbon source, with the glycerol flux to 1,3-bisphosphoglycerate averaging 64% (*Tomàs-Gamisans et al., 2019*).

Discrete $\mu$ values showed decreases over the culture time of 2.1- and 4.4-times lower on average from 6 to 12 and from 12 to 18 h of culture, respectively (Fig. 1B). Hence, the discrete $\mu$ values can be classified as high, medium, and low $\mu$ (*i.e.*, 0.253, 0.123, and 0.030 h$^{-1}$ on average, respectively). Although no significant differences were observed in $\mu$ values during the exponential growth phase, either when comparing the same strain grown in different carbon sources or when assessing the two strains grown under the same carbon source, significant differences were observed in the discrete $\mu$ values between the two strains in the glucose cultures and between the glucose and glycerol cultures for the $P_{GAP}$-$T_{GAP}$-based strain at 12 and 18 h of culture (Fig. 1B).

## *FTEII*- and *GAPDH*-transcript levels and extracellular $Y_{p/x}$ along the culture time

Figures 2A and 2B show the *FTEII*- and *GAPDH*-transcript levels at the three sampling times for glucose-grown (Fig. 2A) or glycerol-grown cells (Fig. 2B) of the two strains.

In the $P_{GAP}$-$T_{AOX1}$-based strain, *FTEII*-transcript levels peaked at 6 h of culture, which coincided with the highest $\mu$ (0.253 h$^{-1}$), regardless of whether the cells were grown in glucose or glycerol (Fig. 1B). The levels then decreased and remained stable from 12 to 18 h of culture. Meanwhile, *GAPDH*-transcript levels showed a similar trend to the *FTEII*-transcript levels until 12 h in the glucose cultures (Fig. 2A), though no significant differences were observed in *GAPDH*-transcript levels from 6 to 12 h in the glycerol cultures (Fig. 2B). At 18 h of culture, when the glucose or glycerol was almost depleted, the

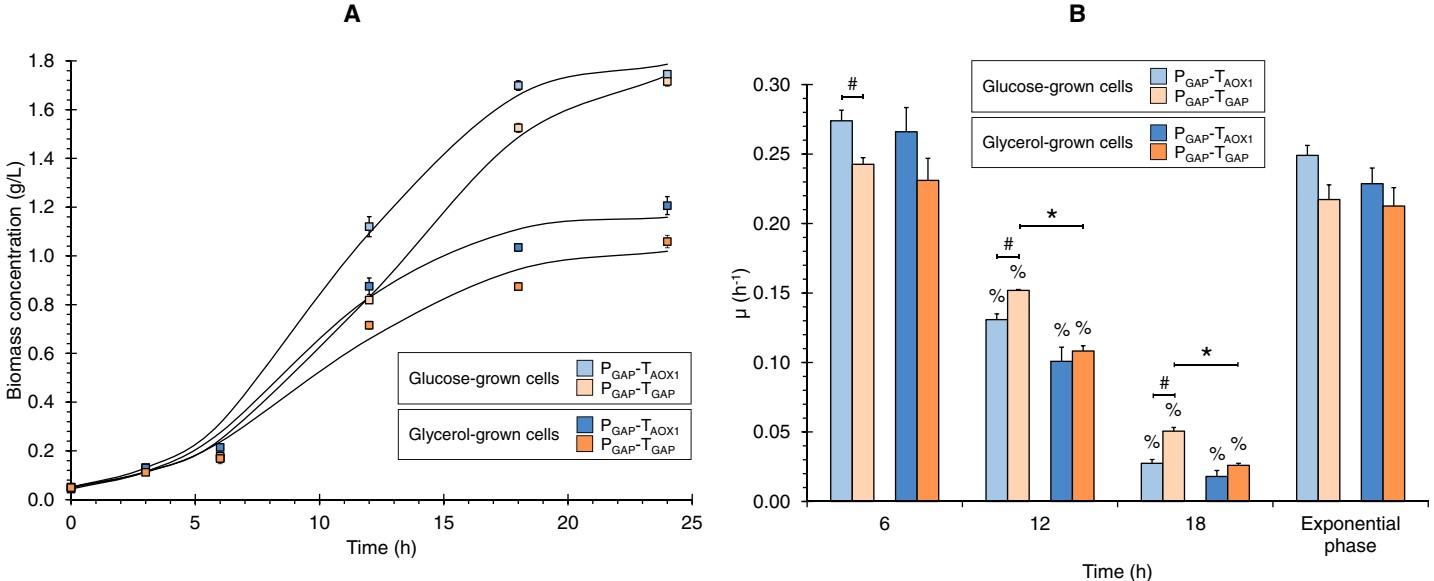

**Figure 1** (A) Growth kinetics, and (B) specific growth rate (μ) at different culture times for the $P_{GAP}$-$T_{AOX1}$-based and $P_{GAP}$-$T_{GAP}$-based strains in glucose (BMGlc) and glycerol (BMGly) media. Data is presented as the mean ± standard error from three independent kinetics experiments. Significant differences ($p < 0.05$) between data (%) from two consecutive times of culture, (*) glucose-grown and glycerol-grown cells for the same strain, and (#) the two strains grown in the same carbon source.

*GAPDH* gene exhibited a significant decrease, showing lower transcript levels than those of the *FTEII* gene.

Although the transcriptional activity trend profiles of the *FTEII* and *GAPDH* genes in the glucose-grown $P_{GAP}$-$T_{GAP}$-based cells were similar to those in the $P_{GAP}$-$T_{AOX1}$-based cells up to 12 h of culture, the transcript levels remained unchanged in the glycerol-grown $P_{GAP}$-$T_{GAP}$-based cells from 6 to 12 h of culture. Moreover, in the $P_{GAP}$-$T_{GAP}$-based strain, both genes exhibited a pronounced decrease in transcript levels at 18 h of culture with either carbon source tested.

Along the progression of culture (*i.e.*, decreasing μ), the extracellular $Y_{p/x}$ values consistently increased for the $P_{GAP}$-$T_{AOX1}$-based strain in the two carbon sources tested (Fig. 2C). Similarly, the extracellular $Y_{p/x}$ values for the $P_{GAP}$-$T_{GAP}$-based cell cultures increased from 6 to 12 h in the two carbon sources; however, no significant differences were seen in the extracellular $Y_{p/x}$ from 12 to 18 h in both glucose and glycerol cultures (Fig. 2C). Overall, the *FTEII*-transcript levels and the extracellular $Y_{p/x}$ were related in an inversely proportional manner.

## Comparison of *FTEII*- and *GAPDH*-transcript levels of the same strain grown in the same carbon source

Although both the *FTEII* and *GAPDH* genes were regulated by $P_{GAP}$ in the two strains, the transcript levels of the endogenous *GAPDH* gene for the $P_{GAP}$-$T_{AOX1}$-based strain were higher than those of the *FTEII* gene at 6 and 12 h of culture in glucose (1.7 times higher on average) (Fig. 2A) and at 12 h of culture in glycerol (2.0 times higher) (Fig. 2B). Nevertheless, *FTEII* exhibited on average 2.1 times higher transcript levels, compared to
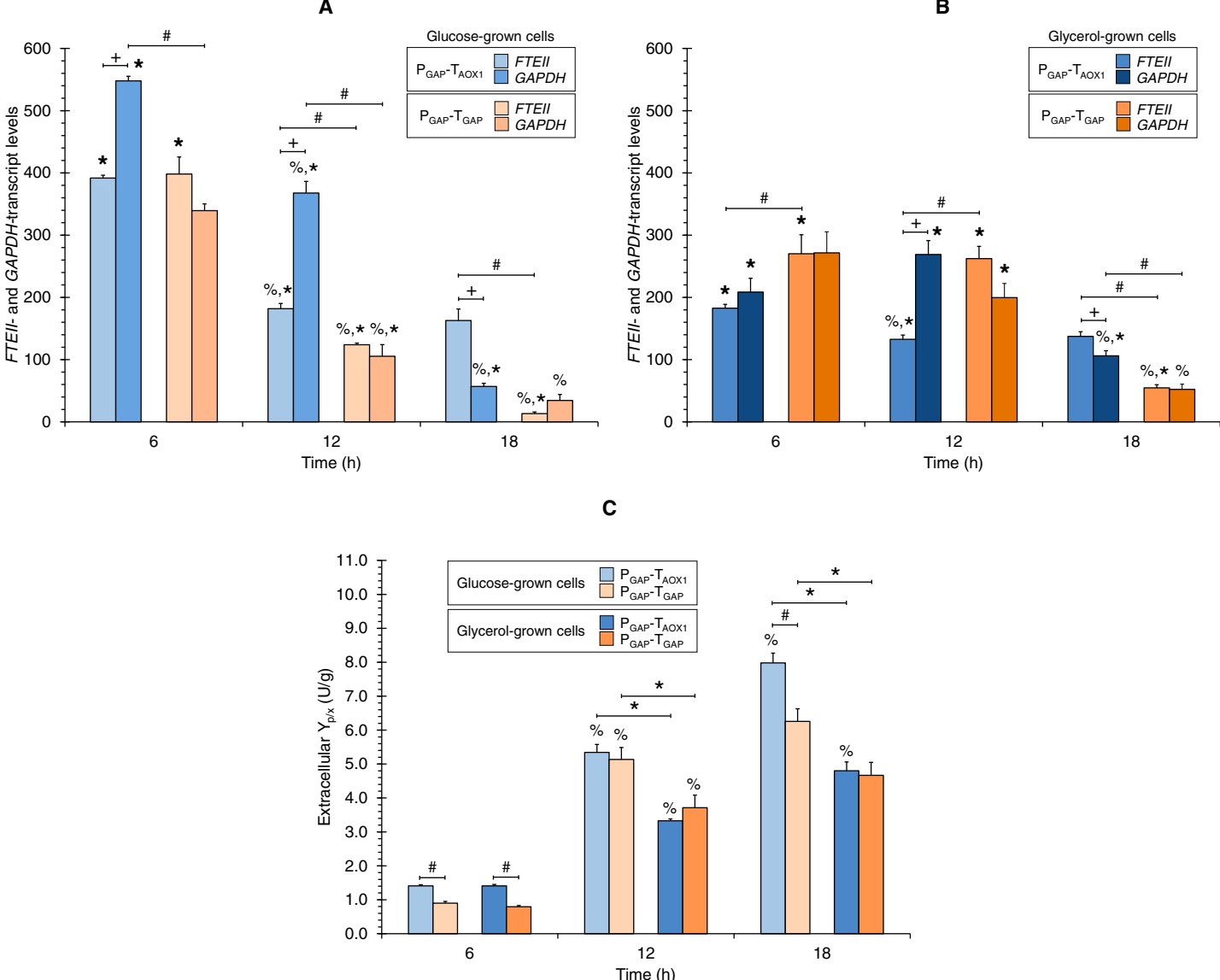

**Figure 2** (A) *FTEII*- and *GAPDH*-transcript levels for the $P_{GAP}$-based strains in glucose and (B) glycerol media. (C) Extracellular $Y_{p/x}$ for cultures of the $P_{GAP}$-based strains in glucose and glycerol media. Data is presented as the mean ± standard error from three independent cultures. Significant differences ($p < 0.05$) between data (%) from two consecutive times of culture, (+) *FTEII*-transcript and *GAPDH*-transcript levels of the same strain grown in the same carbon source, (*) glucose-grown and glycerol-grown cells for the same strain, and (#) the two strains grown in the same carbon source.

*GAPDH* at 18 h of culture in the two carbon sources (Figs. 2A and 2B). No significant differences were observed between *FTEII*- and *GAPDH*-transcript levels for the $P_{GAP}$-$T_{AOX1}$-based strain at 6 h of culture in glycerol (Fig. 2B).

Conversely, no significant differences in transcript levels were seen in the $P_{GAP}$-$T_{GAP}$-based strain between the heterologous *FTEII* and endogenous *GAPDH* genes at the three sampling times of culture in any of the carbon sources.

## Carbon source effect on *FTEII*- and *GAPDH*-transcript levels

The *FTEII*- and *GAPDH*-transcript levels for the $P_{GAP}$-$T_{AOX1}$-based strain were higher in glucose-grown cells (Fig. 2A) compared to glycerol-grown cells (Fig. 2B) at 6 h (2.2 and 2.6 times, respectively) and at 12 h of culture (1.4 times for the two genes). Nevertheless, no significant differences were observed in *FTEII*-transcript levels at 18 h of culture between cells grown in glucose or glycerol, while *GAPDH*-transcript levels were 1.9 times higher in glycerol-grown cells compared to glucose-grown cells.

Although the *FTEII*- and *GAPDH*-transcript levels for the $P_{GAP}$-$T_{AOX1}$-based cells were higher in glucose (Fig. 2A) than in glycerol (Fig. 2B) cultures during the exponential growth phase (high and medium μ), *FTEII*- and *GAPDH*-transcript levels for $P_{GAP}$-$T_{GAP}$-based cells were higher in glycerol (Fig. 2A), rather than in glucose (Fig. 2B) cultures most of the time. Specifically, $P_{GAP}$-$T_{GAP}$-based cells exhibited higher *FTEII*-transcript levels in glycerol than in glucose cultures at 12 and 18 h of culture (2.1 and 4.1 times, respectively), while at 6 h of culture, these cells showed higher *FTEII*-transcript levels in glucose (1.5 times) than in glycerol cultures. The *GAPDH*-transcript levels were 1.9 times higher in glycerol-grown rather than in glucose-grown cells at 12 h of culture, though no significant differences were seen at 6 and 18 h of culture.

## Comparison of transcript levels between strains grown in the same carbon source

No significant differences were observed in the *FTEII*-transcript levels at 6 h of culture between the two strains grown in glucose (Fig. 2A), though the *FTEII*-transcript levels were higher for the $P_{GAP}$-$T_{AOX1}$-based cells at 12 h (1.5 times) and 18 h (12.3 times) of culture, compared to $P_{GAP}$-$T_{GAP}$-based cells that were either grown in glucose or glycerol at 18 h (2.5 times) of culture (Figs. 2A and 2B). Nevertheless, *FTEII*-transcript levels at 6 and 12 h of culture for $P_{GAP}$-$T_{GAP}$-based cells grown in glycerol (Fig. 2B) were 1.5 and 2.0 times higher, respectively, compared to $P_{GAP}$-$T_{AOX1}$-based cells also grown in glycerol.

The *GAPDH*-transcript levels for the $P_{GAP}$-$T_{AOX1}$-based cells were higher compared to the $P_{GAP}$-$T_{GAP}$-based cells at 6 and 12 h of culture (1.6 and 3.5 times, respectively) in glucose cultures (Fig. 2A) and at 18 h (2.0 times) of culture in glycerol (Fig. 2B). No significant differences in *GAPDH*-transcript levels were observed between the two strains grown in glucose at 18 h of culture or in glycerol at 6 and 12 h of culture.

## Comparison of extracellular $Y_{p/x}$ between strains and the effect of carbon source

The $P_{GAP}$-$T_{AOX1}$-based cell cultures in both carbon sources showed an average extracellular $Y_{p/x}$ value that was 1.3 times greater than that of the $P_{GAP}$-$T_{GAP}$-based cell cultures (Fig. 2C). Moreover, no significant differences were observed in the extracellular $Y_{p/x}$ between the glucose and glycerol cultures at 6 h of culture for either of the two strains. On average, the extracellular $Y_{p/x}$ was 1.6 times higher in glucose compared to glycerol cultures at 12 and 18 h of culture for the $P_{GAP}$-$T_{AOX1}$-based strain, and 1.4 times higher for the $P_{GAP}$-$T_{GAP}$-based strain.

### Comparison of *FTEII*-transcript levels ratios and extracellular $Y_{p/x}$ ratios

The *FTEII*-transcript levels inter-strain ratios and extracellular $Y_{p/x}$ inter-strain ratios were different, and also different in terms of the culture time (Fig. 3A). The extracellular $Y_{p/x}$ inter-strain ratios were higher than *FTEII*-transcript levels inter-strain ratios at 6 h of culture in the tested carbon sources. In contrast, at the medium μ (0.123 h$^{-1}$), the extracellular $Y_{p/x}$ inter-strain ratio was higher than the *FTEII*-transcript levels inter-strain ratio only in glycerol cultures, while at 18 h of culture, the *FTEII*-transcript levels inter-strain ratio was higher than the extracellular $Y_{p/x}$ inter-strain ratio in both carbon sources. These results suggest that the post-transcriptional activity was impacted by the presence of $T_{AOX1}$ within the heterologous gene structure and was dependent upon the carbon source and the stage of cell growth.

The ratios of the *FTEII*-transcript levels and the extracellular $Y_{p/x}$ ratios from the glucose-grown to glycerol-grown cells were different, and also different in terms of the culture time (Fig. 3B). At 6 h of culture, the *FTEII*-transcript levels carbon-source ratio for the $P_{GAP}$-$T_{AOX1}$-based strain was higher than the corresponding extracellular $Y_{p/x}$ ratio, while significant differences were not observed between the two ratios for the $P_{GAP}$-$T_{GAP}$-based cell cultures. Moreover, no significant differences were observed between the two ratios at 12 h of culture for the $P_{GAP}$-$T_{AOX1}$-based strain. In contrast, at 12 h of culture for the $P_{GAP}$-$T_{GAP}$-based strain and at 18 h of culture for the two strains, the extracellular $Y_{p/x}$ carbon-source ratios were higher than the corresponding *FTEII*-transcript levels ratios.

## DISCUSSION

A methanol-free system would be preferable for recombinant protein production in large-scale bioreactors due to the critical safety concerns associated with its handling and storage (*Mattanovich et al., 2014*). Although $P_{GAP}$-based *K. phaffii* strains offer a viable solution for protein production in a methanol-free system (*García-Ortega et al., 2019*), and the production levels in a *K. phaffii* expression system are primarily dictated by the combined action of the promoter and the transcriptional terminator (*Matsuyama, 2019*), little is known about the $P_{GAP}$ transcriptional activity on the heterologous gene in combination with the $T_{GAP}$, compared to $P_{GAP}$ in combination with the $T_{AOX1}$ and its effect on the extracellular $Y_{p/x}$. Therefore, we characterized the transcriptional activity of the two combinations of regulatory DNA elements, $P_{GAP}$-$T_{AOX1}$ and $P_{GAP}$-$T_{GAP}$, at different stages of cell growth and with two carbon sources (glucose and glycerol) to evaluate the effect of the regulatory element combinations along with the μ and carbon source on the heterologous *FTEII*- and endogenous *GAPDH*-transcript levels and the extracellular $Y_{p/x}$. The results indicate that the *FTEII*-transcript and *GAPDH*-transcript levels and the extracellular $Y_{p/x}$ values were affected by the combinations of regulatory DNA element, $P_{GAP}$-$T_{AOX1}$ or $P_{GAP}$-$T_{GAP}$, within the heterologous gene structure, the carbon source, and the stage of cell growth. Moreover, the heterologous *FTEII*-transcript and endogenous *GAPDH*-transcript levels tended to decrease during the culture time, while the extracellular $Y_{p/x}$ values tended to increase.

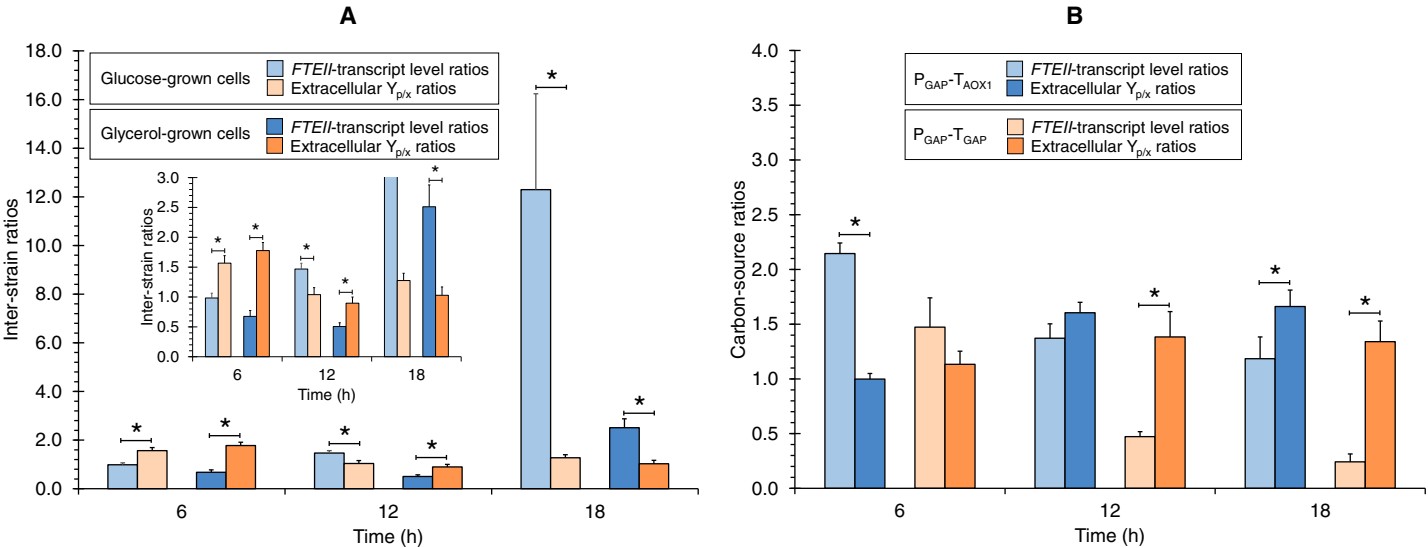

**Figure 3** *FTEII*-transcript level ratios and extracellular $Y_{p/x}$ ratios for (A) $P_{GAP}$-$T_{AOX1}$-based *vs.* $P_{GAP}$-$T_{GAP}$-based strains (inter-strain ratios) and for (B) glucose *vs.* glycerol cultures (carbon-source ratios). Data is presented as the mean ± standard error from three independent cultures. An asterisk (*) indicates significant differences ($p < 0.05$) between inter-strain ratios from cultures in the same carbon source or carbon-source ratios from cultures of the same strain.

Terminator sequences comprise a 3′UTR followed by a necessary region for ending mRNA transcription. Therefore, as a first approach for deciding the $T_{GAP}$ sequence, we performed a RNA mapping analysis to detect a subregion within the inter-CDS region of the *GAPDH* gene and the downstream gene that lacked aligned RNA reads. Nevertheless, the RNA reads were mapped in the entire inter-CDS region. The presence of transcripts beyond the 3′UTR of the *GAPDH* gene could be due to the presence of cryptic unstable transcripts (CUTs) and/or stable uncharacterized transcripts (SUTs) (*Marquardt, Hazelbaker & Buratowski, 2011*). CUTs are a subset of non-coding RNAs produced from intergenic and intragenic regions and have been described as the principal type of RNA polymerase II transcripts in *S. cerevisiae* (*Wyers et al., 2005*). SUTs are similar to CUTs, though they are less susceptible to degradation than CUTs. Strongly expressed CUTs are enriched for genes involved in glucose catabolism (*Neil et al., 2009*), like the *GAPDH* gene. Hence, we used the 3′RACE technique to define the 3′UTR sequence of the *GAPDH* gene and considered this 3′UTR sequence with an additional 50 downstream nucleotides as the $T_{GAP}$ sequence for evaluation, in accordance with known recommendations for yeast transcriptional terminators (*Curran et al., 2013*). The identified $T_{GAP}$ sequence has the characteristic architecture of the yeast transcriptional terminators with four A + T-rich elements (*Graber, McAllister & Smith, 2002*; *Graber, 2003*).

We chose single-copy clones from the engineered strains ($P_{GAP}$-$T_{AOX1}$-based and $P_{GAP}$-$T_{GAP}$-based) to prevent the influence of gene dosage on the heterologous gene-transcript levels and the extracellular $Y_{p/x}$, since strains with multiple copies of the expression cassette tend to exhibit higher heterologous gene-transcript and protein levels than those with a single copy (*Cereghino & Cregg, 2000*; *Looser et al., 2015*; *Mombeni et al., 2020*). This

deliberate choice of single-copy clones allowed for a more controlled assessment, by isolating the influence of gene dosage variability on the measured outcomes.

Before conducting the in-depth functionality analysis of the regulatory DNA element combinations (*i.e.*, $P_{GAP}$-$T_{AOX1}$ and $P_{GAP}$-$T_{GAP}$ pairs) in the two single-copy strains, we initially confirmed the functionality of the promoter-terminator pair by detecting *FTEII*-transcript and FTEII protein, along with phytase enzyme activity.

Although an intracellular green fluorescent protein is usually used as the reporter protein to measure promoter strength (*Hartner et al., 2008*; *Qin et al., 2011*; *Prielhofer et al., 2013*), this approach measures the transcription together with the translation process and does not evaluate the protein secretion process, even though the *K. phaffii* expression system is often used for the extracellular production of recombinant proteins. Therefore, we decided to assess the heterologous gene-transcript levels by RT-qPCR and the extracellular $Y_{p/x}$ using a heterologous reporter gene coding for a protein secreted into the culture medium. This approach is similar to what we previously used to study the *K. phaffii 1033* promoter and transcriptional terminator, which exhibits constitutive, weak, non-methanol-dependent transcriptional activity (*Robainas-del-Pino et al., 2023*). Moreover, this approach is useful for evaluating not only the effects of the promoter and the transcriptional terminator on the heterologous-gene transcription and transcript stability, but also for assessing the regulatory element's influence on translation and protein secretion.

*FTEII*- and *GAPDH*-transcriptional activity profiles showed different patterns in response to carbon sources, μ values, and the $P_{GAP}$-transcriptional terminator combination present within the heterologous gene structure. Specifically, the $P_{GAP}$-$T_{AOX1}$-based cells exhibited distinct transcriptional patterns for the *FTEII* and *GAPDH* genes when grown in glucose or glycerol. These results indicate a probable competitive regulation of the transcriptional activity between the two genes due to the presence of $T_{AOX1}$ in the *FTEII* gene structure. This competitive regulation could lead to differences in cis- and/or trans-regulation mechanisms, and to higher transcript levels of the endogenous *GAPDH* gene compared to the heterologous *FTEII* gene at 6 and 12 h (high and medium μ values, respectively) in both glucose and glycerol cultures. A similar regulation competition between a heterologous gene and *GAPDH* gene expression using different combinations of $P_{GAP}$-transcriptional terminators has been reported previously (*Dou et al., 2021*; *Robainas-del-Pino et al., 2023*).

The higher transcript levels for the two genes observed in glucose-grown compared to glycerol-grown cells during the exponential growth phase (6 and 12 h of culture) in the $P_{GAP}$-$T_{AOX1}$-based strain are consistent with previous reports where the $P_{GAP}$ transcriptional activity in glucose-grown cells was greater than that in glycerol-grown cells (*Waterham et al., 1997*).

In the $P_{GAP}$-$T_{AOX1}$-based strain, when glucose or glycerol was becoming depleted and the μ decreased, *FTEII* and *GAPDH* genes exhibited significant downregulation. This transcriptional pattern in both genes was likely triggered by carbon source depletion. Nevertheless, the higher transcript levels of *FTEII* compared to *GAPDH* in both glucose-grown and glycerol-grown cells at 18 h of culture suggest that the downregulation

due to carbon source depletion was stronger in the presence of $T_{GAP}$ compared to $T_{AOX1}$. Moreover, the higher *GAPDH*-transcript levels at 18 h of culture in glycerol compared to glucose cultures indicate that the downregulation due to carbon source depletion in the presence of $T_{GAP}$ was even stronger in glucose than in glycerol cultures. These findings indicate a different metabolic downregulation mechanism and/or higher mRNA stability due to the presence of $T_{AOX1}$ (*Ramakrishnan et al., 2020*; *Ito et al., 2020*; *Herrera-Estala et al., 2022*). Therefore, the metabolic downregulation of the two genes was dependent on the $P_{GAP}$-transcriptional terminator combination within the gene structure and on the carbon source.

When the heterologous *FTEII* and endogenous *GAPDH* genes shared the same promoter-terminator combination in their gene structures (*i.e.*, the $P_{GAP}$-$T_{GAP}$-based strain) a comparable transcriptional activity of the two genes was observed in glucose and glycerol cultures. These findings support the idea that the presence of $T_{GAP}$ in heterologous *FTEII* and the *GAPDH* gene structures triggered a balanced competition in their transcriptional regulation *via* cis- and trans-regulatory elements. The *FTEII*- and *GAPDH*-transcript levels exhibited a dynamic response to the μ and the carbon sources. The highest transcript levels were reached in glucose cultures at a high μ ($0.253 \ h^{-1}$). Nevertheless, the transcript levels were higher in glycerol cultures compared to glucose cultures when the μ decreased (at 12 and 18 h of culture). These findings could be related to the presence of $T_{GAP}$ in both genes, causing gene regulation that depends not only on the carbon source but also on the μ value. Like the $P_{GAP}$-$T_{AOX1}$-based cells, the $P_{GAP}$-$T_{GAP}$-based cells also showed downregulation of the two genes as the μ decreased and carbon sources were being depleted. The metabolic downregulation was more pronounced in glucose-grown cells, resulting in lower transcript levels of both genes compared to glycerol-grown cells at 18 h of culture.

This investigation revealed distinct patterns in *FTEII* transcriptional activity under the regulation control of $P_{GAP}$ between the $P_{GAP}$-$T_{AOX1}$-based and $P_{GAP}$-$T_{GAP}$-based strains over the culture time in glucose and glycerol. These transcriptional activity profiles were influenced by the $P_{GAP}$-transcriptional terminator combination, carbon sources, and cell growth stages (specific growth rate). The lower *FTEII*-transcript levels observed in the $P_{GAP}$-$T_{GAP}$-based cells at 12 and 18 h of culture, compared to those in the $P_{GAP}$-$T_{AOX1}$-based cells, both cultured in glucose, were likely due to stronger downregulation induced by decreasing glucose levels in the presence of $T_{GAP}$ or to an increased stability of *FTEII* transcripts conferred by $T_{AOX1}$. Consequently, the *FTEII*-transcript levels remained unchanged from 12 to 18 h of culture in the $P_{GAP}$-$T_{AOX1}$-based strain. Conversely, the higher *FTEII*-transcript levels observed in the $P_{GAP}$-$T_{GAP}$-based cells at 6 and 12 h of culture in glycerol, compared to the $P_{GAP}$-$T_{AOX1}$-based cells, suggest enhanced transcriptional activity of $P_{GAP}$ when $T_{GAP}$ was present in the *FTEII* gene structure compared to $T_{AOX1}$ when glycerol was used as the carbon source. Nevertheless, the $P_{GAP}$-$T_{AOX1}$-based strain exhibited the highest *FTEII*-transcript levels at 18 h in glycerol, suggesting stronger downregulation due to glycerol depletion in the presence of $T_{GAP}$ as the transcriptional terminator in the *FTEII* gene structure in the $P_{GAP}$-$T_{GAP}$-based strain or to an increased stability of *FTEII* transcripts conferred by $T_{AOX1}$.

The transcriptional activity of the *GAPDH* gene in the glucose-grown $P_{GAP}$-$T_{GAP}$-based cells had a pattern that was similar to that of the $P_{GAP}$-$T_{AOX1}$-based strain. The lower transcript levels of the *GAPDH* gene in the $P_{GAP}$-$T_{GAP}$-based cells during the cell growth stage of high and medium $\mu$ (0.253 and 0.123 h$^{-1}$, respectively), compared to the $P_{GAP}$-$T_{AOX1}$-based cells, supports the idea that the presence of $T_{AOX1}$ in the *FTEII* gene structure induced a distinct regulation of the *GAPDH* gene, resulting in increased transcriptional activity in glucose-grown cells of the $P_{GAP}$-$T_{AOX1}$-based strain. This effect was not observed in glycerol cultures, suggesting that the presence of $T_{AOX1}$ in the *FTEII* gene structure did not enhance the transcriptional activity of the *GAPDH* gene at high $\mu$ (0.253 h$^{-1}$) when glycerol was used as the carbon source. Furthermore, *GAPDH*-transcript levels for the two strains were comparable at 6 and 12 h of culture in glycerol, but higher for $P_{GAP}$-$T_{AOX1}$-based cells at 18 h in glycerol compared to the $P_{GAP}$-$T_{GAP}$-based cells. The *GAPDH*-transcript levels were downregulated in both strains as the carbon source was being depleted between 12 and 18 h of culture. The metabolic downregulation was more pronounced in the $P_{GAP}$-$T_{AOX1}$-based cells grown in glucose, but more pronounced for the $P_{GAP}$-$T_{GAP}$-based strain grown in glycerol. Taken together, these findings indicate that the specific $P_{GAP}$-terminator combination within the heterologous gene structure, the carbon source, and the stage of cell growth influence the regulation of *GAPDH* gene transcription.

In general, the extracellular $Y_{p/x}$ values were higher in the $P_{GAP}$-$T_{AOX1}$-based strain compared to the $P_{GAP}$-$T_{GAP}$-based strain. These extracellular $Y_{p/x}$ values showed an increasing trend as the culture progressed for the $P_{GAP}$-$T_{AOX1}$-based strain and up to the 12 h of culture for the $P_{GAP}$-$T_{GAP}$-based strain, with higher values in glucose than in glycerol cultures. The results revealed a direct proportional relationship between $P_{GAP}$-driven *FTEII*-transcript levels and $\mu$ values in most cases, while an inverse proportional relationship was seen between extracellular $Y_{p/x}$ and $\mu$ values. Consequently, an inverse proportional relationship was seen between *FTEII*-transcript levels and extracellular $Y_{p/x}$ values. These findings are in agreement with previous reports of higher $P_{GAP}$ transcriptional activity at high $\mu$ (*Looser et al., 2015*), and with our previous results (*Herrera-Estala et al., 2022*) where we showed that the extracellular $Y_{p/x}$ values for the same heterologous protein (*i.e.*, phytase FTEII) under the control of $P_{GAP}$ and $T_{AOX1}$ were highest under conditions of low $\mu$ and low heterologous-gene transcript levels. These results further confirm our previous hypothesis that the proper folding of the heterologous protein and/or post-translational processing within the secretory pathway represents the rate-limiting step for achieving high yields of extracellular proteins (*Herrera-Estala et al., 2022*).

The comparison of *FTEII*-transcript level ratios for $P_{GAP}$-$T_{AOX1}$-based *vs.* $P_{GAP}$-$T_{GAP}$-based strains grown in the same carbon source and the corresponding extracellular $Y_{p/x}$ inter-strain ratios provides insight into the effect of the transcriptional terminator within the heterologous gene structure on post-transcriptional cellular activity for the heterologous protein (*i.e.*, translation and/or heterologous protein secretion pathway activity). Extracellular $Y_{p/x}$ inter-strain ratios higher than their corresponding *FTEII*-transcript level inter-strain ratios indicate that $T_{AOX1}$ enhanced the post-transcriptional activity compared to $T_{GAP}$; otherwise, $T_{GAP}$ enhanced post-transcriptional activity

compared to $T_{AOX1}$. The absence of differences between the two ratios indicates a comparable terminator effect on post-transcriptional activity.

Our findings reveal differences between the two inter-strain ratios, indicating that the transcriptional terminator within the *FTEII* gene structure played a pivotal role in influencing translation and/or heterologous protein secretion, as previously described for other transcriptional terminators (*Mayr, 2019*; *Kuersten & Goodwin, 2003*). Furthermore, the findings indicate that the presence of $T_{AOX1}$ within the heterologous gene structure significantly enhanced translation and/or heterologous protein secretion in glucose cultures at a high μ (0.253 h$^{-1}$), as well as in glycerol cultures at both high and medium μ (0.253 and 0.123 h$^{-1}$, respectively), compared to $T_{GAP}$. In contrast, $T_{GAP}$ was more effective than $T_{AOX1}$ at enhancing post-transcriptional activity in glucose cultures at a medium μ (0.123 h$^{-1}$) and in cultures with either carbon source tested at a low μ (0.030 h$^{-1}$).

Like the comparison of the inter-strain ratios, the comparison between *FTEII*-transcript level ratios for glucose-grown and glycerol-grown cells and the corresponding extracellular $Y_{p/x}$ carbon-source ratios offers insight into the effect of the carbon source on post-transcriptional cellular activity for the heterologous protein (*i.e.*, translation and/or heterologous protein secretion pathway activity). Differences were observed between the two carbon-source ratios, indicating that post-transcriptional activity was influenced by the tested carbon source. The findings indicate that glycerol enhanced the post-transcriptional activity of the heterologous protein compared to glucose in cultures at a high μ (0.253 h$^{-1}$) when the $P_{GAP}$-$T_{AOX1}$ combination was present within the heterologous gene structure, as we previously observed with another promoter-terminator pair using the same reporter gene (*Robainas-del-Pino et al., 2023*). In contrast, glucose enhanced the post-transcriptional activity of the heterologous protein compared to glycerol at a low μ (0.030 h$^{-1}$) when the $P_{GAP}$-$T_{AOX1}$ combination was present and at medium and low μ (0.123 and 0.030 h$^{-1}$, respectively) when the $P_{GAP}$-$T_{GAP}$ combination was present within the heterologous gene structure.

## CONCLUSIONS

The findings provide crucial insights into gene transcription modulation and its impact on extracellular $Y_{p/x}$ in a $P_{GAP}$-based *K. phaffii* strain for the extracellular production of a heterologous protein. The combination of $P_{GAP}$-transcriptional terminator ($T_{AOX1}$ or $T_{GAP}$) influenced the transcription of both the heterologous and endogenous *GAPDH* genes, extracellular $Y_{p/x}$ values, and translation and/or heterologous protein secretion, with the carbon source and the stage of cell growth acting as modulators. In terms of increasing extracellular $Y_{p/x}$ in a $P_{GAP}$-based *K. phaffii* system, the $T_{AOX1}$ proved to be a more suitable transcriptional terminator than the $T_{GAP}$, principally when the culture is carried out at a low μ (0.030 h$^{-1}$) using glucose as the carbon source, even though the preference for glucose or glycerol might shift in a strain harboring multiple copies of the expression cassette containing $T_{AOX1}$. The optimization of regulatory elements and growth conditions presents opportunities for enhancing the production of biomolecules of interest. The implications of these findings extend to metabolic engineering strategies and the fine-tuning of bioprocesses for increased efficiency.

## ACKNOWLEDGEMENTS

The authors thank Jesus Uxue Mendoza Leyva and Giselle Cao Luna for their technical support and Glen D. Wheeler for his stylistic suggestions in the preparation of the manuscript.

### Funding

This work was supported by the Consejo Nacional de Ciencia y Tecnología (CONACYT), Mexico (Grant number CB-2016-286093). Nancy Pentón-Piña and Yanelis Robainas-del-Pino received fellowships from CONACYT. The funders had no role in study design, data collection and analysis, decision to publish, or preparation of the manuscript.

### Grant Disclosures

The following grant information was disclosed by the authors:
Consejo Nacional de Ciencia y Tecnología (CONACYT), Mexico: CB-2016-286093.

### Competing Interests

The authors declare that they have no competing interests.

### Author Contributions

- José M. Viader-Salvadó conceived and designed the experiments, analyzed the data, prepared figures and/or tables, authored or reviewed drafts of the article, and approved the final draft.
- Nancy Pentón-Piña conceived and designed the experiments, performed the experiments, analyzed the data, prepared figures and/or tables, authored or reviewed drafts of the article, and approved the final draft.
- Yanelis Robainas-del-Pino performed the experiments, authored or reviewed drafts of the article, and approved the final draft.
- José A. Fuentes-Garibay performed the experiments, authored or reviewed drafts of the article, and approved the final draft.
- Martha Guerrero-Olazarán conceived and designed the experiments, analyzed the data, prepared figures and/or tables, authored or reviewed drafts of the article, and approved the final draft.

### Data Availability

    The raw measurements are available in the Supplemental File.

### Supplemental Information

Supplemental information for this article can be found online at http://dx.doi.org/10.7717/peerj.18181#supplemental-information.

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
