# Peer review of "Effect of AOX1 and GAP transcriptional terminators on transcript levels of both the heterologous and the GAPDH genes and the extracellular Yp/x in GAP promoter-based Komagataella phaffii strains"

_PeerJ, doi:10.7717/peerj.18181_

## Round 0.1 · original submission · Major Revisions

· Academic Editor

Major Revisions

Three experts assessed your manuscript and found the content interesting and suitable for publication after addressing several concerns. Please note that two of the reviewers agree that additional experimentation with the system under methanol induction is required. Moreover, concerns about the different vectors included in the study are also raised.

Reviewer 1 ·

Basic reporting

Overall, the research work is carried out appropriately. Raw data and supplementary material were provided.
The work has the necessary references and the approach taken to answer the research questions is appropriate.

Experimental design

The methods used to answer the questions are adequate, and well described for possible reproductions. However, I consider that if one of the objectives of this work is to avoid the use of methanol for the production tests of heterologous proteins in this microorganism, it was necessary to do a production test using the basic system (using methanol) and thus compare the production results, which will give the present work much more solidity.

Validity of the findings

The conclusions shown are supported by the results shown.

Additional comments

The abstract section is too long, I suggest you trim it down a bit.

Line 73, 276, 278: K. phaffii

line 190: 16, 000 x g

line 192. Reference



Regarding the production cost for the growth of the microorganism in glucose or glycerol, is it correlated with the production of the heterologous protein, and what is invested in the cultivation?



I consider it important to perform an activity test with the heterologous enzyme produced; this is to conclude that the production is effective and the activity is not affected.


I suggest that the conclusion be more concise, it is very long.

Reviewer 2 ·

Basic reporting

In this manuscript, the authors present the effect of the transcriptional terminators TAOX1 or TGAP on the expression of recombinant proteins in the yeast model Pichia pastoris, in combination with the strong and constitutive promoter from the GAPDH gene (PGAP). They show that the terminator being used within the recombinant gene sequence, along with the carbon source and time of growth, affects different elements of the protein expression, such as the reporter gene and GAPDH transcription levels, extracellular product/biomass yield, and secretion of the recombinant protein.
In general, the english language is good. However, some sections in the results will need to be rewritten, since the way they are presented is confusing, repetitive, and complicated to understand. In addition, the following comments and questions will need to be address for the paper publication:

Abstract
1. Lines 55-56: this low specific growth rate is for the TAOX1 system or for the TGAP? it is not clear. If its for the TAOX1, why low? if above is mentioned that the heterologous expression activity is enhanced at a high specific growth rate.

Introduction
1. Line 99: which terminators are usually used in this expression model?
2. Line 105: although already reported elsewhere, I consider important to mention here the sequence of the TAOX1 being used.

Experimental design

The research question is well defined and relevant, and is answered appropriately with the experimental design used. The materials and methods are clear and with enough information to reproduce them.

Validity of the findings

The findings reported in these manuscript are interesting and will surely help to optimize the expression of recombinant proteins in methanol-free systems. However, as already mentioned, some comments and questions need to be addressed:

Results
1. Line 270: I recommend adding a schematic representation of the TGAP total sequence, explaining the source of the 51 and 50 nucleotides that comprise the sequence and its most relevant characteristics.
2. Line 314: how many repetitions were performed?
3. Lines 337-402: the obtained results in this sections are very interesting but are presented in a very complicated way. The text is hard to understand, and the figures do not really help, specially Figure 4, which I consider the most important. Sections “FTEII- and GAPDH-transcript levels and extracellular Yp/x along the culture time, Comparison of FTEII- and GAPDH-transcript levels, Carbon source effect on FTEII- and GAPDH-transcript levels, and Comparison of transcript levels between strains”, should not be explained separately, since it complicates the understanding of the results and makes the text very repetitive. I ask the authors to rewrite these sections.
4. Line 406: shouldn't it say Figure 4C, instead 4B?
5. Lines 412-430 (Comparison of FTEII-transcript levels ratios and extracellular Yp/x ratios): the only figure mentioned in this section, Figure 4D (line 423), is wrong, I believe it should be Figure 5. Also, all of the figures should be mentioned as the results are being explained, in order to make it easier for the reader.
6. Line 416: which is the medium u? 12 h?
7. I would considered interestingly, besides important, to at least compare the extracellular product/biomass yield of the two promoters systems to what has been reported in the methanol systems. This will add importance to the findings here reported.

Discussion
1. Lines 456-457: this should be mentioned in the methodology
2. Lines 485-486: even when working with a cytosolic protein, shouldn't the protein be secreted when using this plasmid?
3. Line 552: downregulation of who?

Conclusions
1. Line 643: the optimization conditions for the TAOX1 system should be mentioned here. Also, it is important to mention the difference in protein yield between this methanol free system and methanol dependent systems

Additional comments

Some figures will also need to be corrected:
Figure 1: correct the title as suggested (Schematic representation of the pPGAP-FTEII-TAOX1 and pPGAP-FTEII-TGAP expression vectors).
Figure 2: “proteins tested without and with Endo Hf glycosidase” needs to be corrected, since proteins without the glycosidase treatment are not being treated.
Figure 3B: the statistical analysis is missing.
Figure 4: the title is incomplete. The statistical differences are very hard to understand, since is not clear what's being compared with what. I'm aware that the statistical differences are mentioned in the figure description, but, figures are supposed to be understandable without the need of reading the text, and even when reading the text and the figure description, I had a very hard time understanding the results presented in this figure.

Annotated reviews are not available for download in order to protect the identity of reviewers who chose to remain anonymous.

Reviewer 3 ·

Basic reporting

Viader-Salvado et al. report constructing a protein expression system using Pgap as a promoter and the gapdh gene terminator (Tgap) as a transcription terminator. They evaluated the effect of combining the Paox1 and Pgap promoters with the Tgap and Taox1 terminators on heterologous protein production. Overall, it is an interesting study. The structure of the article could be improved.
The abstract and elsewhere in the article state that the Taox1 terminator increases messenger stability (lines 49-51), but no reference is given where this has been proven. Likewise, the only thing measured in the article is the promoter strength; no studies of messenger stability have been shown, so it is very speculative and risky to assert the fact as mentioned earlier.
The introduction is very short and needs to address essential topics for the study. The introduction states that expression from the Pgap promoter is strong and constitutive. However, the results show that the expression of the gapdh gene is not constitutive (it is regulated) (lines 81-82; 386-387). For this reason, it is necessary to address the factors and conditions to which the Pgap promoter responds. It is important to consider whether using glycolytic and gluconeogenic carbon sources affects the activity of Pgad. Also, for this reason, it is important to introduce the metabolism of the carbon sources used in the study in this section.
The explanation given in lines 326-327 needs to be completed; it needs to consider that glucose is a glycolytic carbon source and glycerol is a gluconeogenic source since they continue metabolizing through different pathways.
The different expression patterns of the gapdh and fteII genes mentioned in lines (497-500) of the discussion and in the results section could be because the expression of the gapdh gene is being measured in its wild genomic context, and that the expression plasmid used to express ftelII is not necessarily complete. For comparative purposes, it would be desirable to measure the expression of a gapdh allele cloned in the expression vector being tested, or, if appropriate, to make this clarification in the discussion of the results.
The paragraph on lines 452-480 shows the results of the analysis of region 3 of the gapdh gene, so it could be shown in the results section.
Figures 1 and 2 are support figures, so I suggest showing them as supplementary figures.

Experimental design

The paper complies with the journal requirements in terms of aims and scope. However, there needs to be more information regarding the reporters used. Was the fteII gene cloned with or without a start codon? Does it have ribosome interaction sites or Kozak sequences, or were the gapdh sequences used? The ftell gene is regulated at the post-transcriptional and/or translational level. Regarding internal control of gapdh expression, it is not a comparable control since it was not cloned in the same vector as fteII. In order to see the expression of this allele, the wild-type chromosomal gene gdph should be deleted first.

Validity of the findings

The conclusions section look at a summary of results. A concise conclusion of no more than five lines is suggested.
In general, the management and discussion of the genetic elements and experimental controls used should be improved (see previous section)

Additional comments

No comment

---

## Round 0.2 · accepted · Accept

· Academic Editor

Accept

Two of the original reviewers assessed the revised version of the manuscript and are pleased with the modifications.

Reviewer 2 ·

Basic reporting

I consider that all of my comments and questions were addressed/answered appropriately. My only comment of this corrected manuscript is that I could not find the legends of the supplementary figures in any of the files provided.

Experimental design

No comment.

Validity of the findings

No comment.

Additional comments

In my opinion, the manuscript is ready to be published. However, the authors need to include the legends of the supplementary figures.

Reviewer 3 ·

Basic reporting

The authors have responded appropriately to the comments and suggestions.

Experimental design

The authors have responded appropriately to the comments and suggestions.

Validity of the findings

The authors have responded appropriately to the comments and suggestions.